# STOCHASTIC NORMALIZED GRADIENT DESCENT WITH MOMENTUM FOR LARGE BATCH TRAINING

## ABSTRACT

Stochastic gradient descent (SGD) and its variants have been the dominating optimization methods in machine learning. Compared with small batch training, SGD with large batch training can better utilize the computational power of current multi-core systems like GPUs and can reduce the number of communication rounds in distributed training. Hence, SGD with large batch training has attracted more and more attention. However, existing empirical results show that large batch training typically leads to a drop of generalization accuracy. As a result, large batch training has also become a challenging topic. In this paper, we propose a novel method, called stochastic normalized gradient descent with momentum (SNGM), for large batch training. We theoretically prove that compared to momentum SGD (MSGD) which is one of the most widely used variants of SGD, SNGM can adopt a larger batch size to converge to the $\epsilon$-stationary point with the same computation complexity (total number of gradient computation). Empirical results on deep learning also show that SNGM can achieve the state-of-the-art accuracy with a large batch size.

## 1 INTRODUCTION

In machine learning, we often need to solve the following empirical risk minimization problem:

$$\min_{\mathbf{w} \in \mathbb{R}^d} F(\mathbf{w}) = \frac{1}{n} \sum_{i=1}^{n} f_i(\mathbf{w}), \tag{1}$$

where $\mathbf{w} \in \mathbb{R}^d$ denotes the model parameter, $n$ denotes the number of training samples, $f_i(\mathbf{w})$ denotes the loss on the $i$th training sample. The problem in (1) can be used to formulate a broad family of machine learning models, such as logistic regression and deep learning models.

Stochastic gradient descent (SGD) Robbins & Monro (1951) and its variants have been the dominating optimization methods for solving (1). SGD and its variants are iterative methods. In the $t$th iteration, these methods randomly choose a subset (also called mini-batch) $\mathcal{I}_t \subset \{1, 2, \ldots, n\}$ and compute the stochastic mini-batch gradient $1/B \sum_{i \in \mathcal{I}_t} \nabla f_i(\mathbf{w}_t)$ for updating the model parameter, where $B = |\mathcal{I}_t|$ is the batch size. Existing works Li et al. (2014b); Yu et al. (2019a) have proved that with the batch size of $B$, SGD and its momentum variant, called momentum SGD (MSGD), achieve a $\mathcal{O}(1/\sqrt{TB})$ convergence rate for smooth non-convex problems, where $T$ is total number of model parameter updates.

With the population of multi-core systems and the easy implementation for data parallelism, many distributed variants of SGD have been proposed, including parallel SGD Li et al. (2014a), decentralized SGD Lian et al. (2017), local SGD Yu et al. (2019b); Lin et al. (2020), local momentum SGD Yu et al. (2019a) and so on. Theoretical results show that all these methods can achieve a $\mathcal{O}(1/\sqrt{TKb})$ convergence rate for smooth non-convex problems. Here, $b$ is the batch size on each worker and $K$ is the number of workers. By setting $Kb = B$, we can observe that the convergence rate of these distributed methods is consistent with that of sequential methods. In distributed settings, a small number of model parameter updates $T$ implies a small synchronize cost and communication cost. Hence, a small $T$ can further speed up the training process. Based on the $\mathcal{O}(1/\sqrt{TKb})$ convergence rate, we can find that if we adopt a larger $b$, the $T$ will be smaller. Hence, large batch training can reduce the number of communication rounds in distributed training. Another benefit of adopting

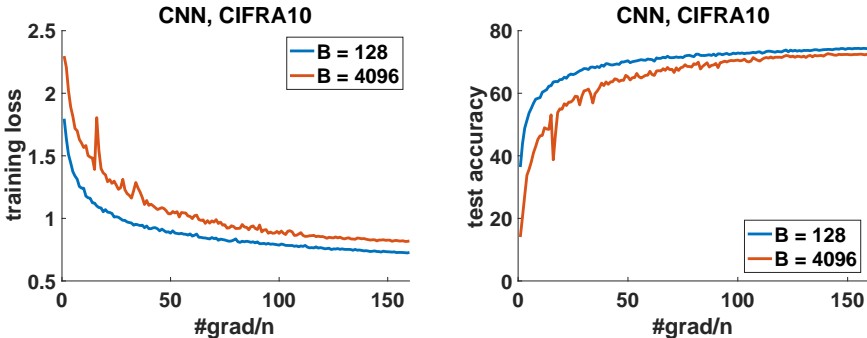

Figure 1: The training loss and test accuracy for training a non-convex model (a network with two convolutional layers) on CIFAR10. The optimization method is MSGD with the poly power learning rate strategy.

large batch training is to better utilize the computational power of current multi-core systems like GPUs You et al. (2017). Hence, large batch training has recently attracted more and more attention in machine learning.

Unfortunately, empirical results LeCun et al. (2012); Keskar et al. (2017) show that existing SGD methods with a large batch size will lead to a drop of generalization accuracy on deep learning models. Figure 1 shows the comparison of training loss and test accuracy between MSGD with a small batch size and MSGD with a large batch size. We can find that large batch training does degrade both training loss and test accuracy. Many works try to explain this phenomenon Keskar et al. (2017); Hoffer et al. (2017). They observe that SGD with a small batch size typically makes the model parameter converge to a flatten minimum while SGD with a large batch size typically makes the model parameter fall into the region of a sharp minimum. And usually, a flatten minimum can achieve better generalization ability than a sharp minimum. Hence, large batch training has also become a challenging topic.

Recently, many methods have been proposed for improving the performance of SGD with a large batch size. The work in Goyal et al. (2017); You et al. (2020) proposes many tricks like warm-up, momentum correction and linearly scaling the learning rate, for large batch training. The work in You et al. (2017) observes that the norms of gradient at different layers of deep neural networks are widely different and the authors propose the layer-wise adaptive rate scaling method (LARS). The work in Ginsburg et al. (2019) also proposes a similar method that updates the model parameter in a layer-wise way. Most of these methods lack theoretical evidence to explain why they can adopt a large batch size. Although the work in You et al. (2020) proposes some theoretical explanations for LARS, the implementation is still not consistent with its theorems in which both of the momentum coefficient and weight decay are set as zeros.

In this paper, we propose a novel method, called stochastic normalized gradient descent with momentum (SNGM), for large batch training. SNGM combines normalized gradient Nesterov (2004); Hazan et al. (2015); Wilson et al. (2019) and Polyak's momentum technique Polyak (1964) together. The main contributions of this paper are outlined as follows:

- We theoretically prove that compared to MSGD which is one of the most widely used variants of SGD, SNGM can adopt a larger batch size to converge to the $\epsilon$-stationary point with the same computation complexity (total number of gradient computation). That is to say, SNGM needs a smaller number of parameter update, and hence has faster training speed than MSGD.

- For a relaxed smooth objective function (see Definition 2), we theoretically show that SNGM can achieve an $\epsilon$-stationary point with a computation complexity of $\mathcal{O}(1/\epsilon^4)$. To the best of our knowledge, this is the first work that analyzes the *computation complexity* of stochastic optimization methods for a relaxed smooth objective function.

- Empirical results on deep learning also show that SNGM can achieve the state-of-the-art accuracy with a large batch size.

## 2 PRELIMINARIES

In this paper, we use $\|\cdot\|$ to denote the Euclidean norm, use $\mathbf{w}^*$ to denote one of the optimal solutions of (1), i.e., $\mathbf{w}^* \in \arg\min_{\mathbf{w}} F(\mathbf{w})$. We call $\mathbf{w}$ an $\epsilon$-stationary point of $F(\mathbf{w})$ if $\|\nabla F(\mathbf{w})\| \leq \epsilon$. The *computation complexity* of an algorithm is the total number of its gradient computation. We also give the following assumption and definitions:

**Assumption 1** ($\sigma$-bounded variance) For any $\mathbf{w}$, $\mathbb{E}\|\nabla f_i(\mathbf{w}) - \nabla F(\mathbf{w})\|^2 \leq \sigma^2$ ($\sigma > 0$).

**Definition 1** (Smoothness) A function $\phi(\cdot)$ is $L$-smooth ($L > 0$) if for any $\mathbf{u}, \mathbf{w}$,

$$\phi(\mathbf{u}) \leq \phi(\mathbf{w}) + \nabla\phi(\mathbf{w})^\top (\mathbf{u} - \mathbf{w}) + \frac{L}{2}\|\mathbf{u} - \mathbf{w}\|^2.$$

$L$ is called *smoothness constant* in this paper.

**Definition 2** (Relaxed smoothness Zhang et al. (2020)) A function $\phi(\cdot)$ is $(L, \lambda)$-smooth ($L \geq 0$, $\lambda \geq 0$) if $\phi(\cdot)$ is twice differentiable and for any $\mathbf{w}$,

$$\|\nabla^2\phi(\mathbf{w})\| \leq L + \lambda\|\nabla\phi(\mathbf{w})\|,$$

where $\nabla^2\phi(\mathbf{w})$ denotes the Hessian matrix of $\phi(\mathbf{w})$.

From the above definition, we can observe that if a function $\phi(\mathbf{w})$ is $(L, 0)$-smooth, then it is a classical $L$-smooth function Nesterov (2004). For a $(L, \lambda)$-smooth function, we have the following property Zhang et al. (2020):

**Lemma 1** If $\phi(\cdot)$ is $(L, \lambda)$-smooth, then for any $\mathbf{u}, \mathbf{w}, \alpha$ such that $\|\mathbf{u} - \mathbf{w}\| \leq \alpha$, we have

$$\|\nabla\phi(\mathbf{u})\| \leq (L\alpha + \|\nabla\phi(\mathbf{w})\|)e^{\lambda\alpha}.$$

All the proofs of lemmas and corollaries of this paper are put in the supplementary.

## 3 RELATIONSHIP BETWEEN SMOOTHNESS CONSTANT AND BATCH SIZE

In this section, we deeply analyze the convergence property of MSGD to find the relationship between smoothness constant and batch size, which provides insightful hint for designing our new method SNGM.

MSGD can be written as follows:

$$\mathbf{v}_{t+1} = \beta\mathbf{v}_t + \mathbf{g}_t, \tag{2}$$
$$\mathbf{w}_{t+1} = \mathbf{w}_t - \eta\mathbf{v}_{t+1}, \tag{3}$$

where $\mathbf{g}_t = 1/B \sum_{i \in \mathcal{I}_t} \nabla f_i(\mathbf{w}_t)$ is a stochastic mini-batch gradient with a batch size of $B$, and $\mathbf{v}_{t+1}$ is the Polyak's momentum Polyak (1964).

We aim to find how large the batch size can be without loss of performance. The convergence rate of MSGD with the batch size $B$ for $L$-smooth functions can be derived from the work in Yu et al. (2019a). That is to say, when $\eta \leq (1 - \beta)^2/((1 + \beta)L)$, we obtain

$$\frac{1}{T}\sum_{t=0}^{T-1}\mathbb{E}\|\nabla F(\mathbf{w}_t)\|^2 \leq \frac{2(1-\beta)[F(\mathbf{w}_0) - F(\mathbf{w}^*)]}{\eta T} + \frac{L\eta\sigma^2}{(1-\beta)^2 B} + \frac{4L^2\eta^2\sigma^2}{(1-\beta)^2},$$

$$= \mathcal{O}(\frac{B}{\eta\mathcal{C}}) + \mathcal{O}(\frac{\eta}{B}) + \mathcal{O}(\eta^2), \tag{4}$$

where $\mathcal{C} = TB$ denotes the *computation complexity* (total number of gradient computation). According to Corollary 1 in Yu et al. (2019a), we set $\eta = \sqrt{B}/\sqrt{T} = B/\sqrt{\mathcal{C}}$ and obtain that

$$\frac{1}{T}\sum_{t=0}^{T-1}\mathbb{E}\|\nabla F(\mathbf{w}_t)\| \leq \sqrt{\mathcal{O}(\frac{1}{\sqrt{\mathcal{C}}}) + \mathcal{O}(\frac{B^2}{\mathcal{C}})}. \tag{5}$$

---

**Algorithm 1** SNGM

---

Initialization: $\eta > 0, \beta \in [0, 1), B > 0, T > 0, \mathbf{u}_0 = \mathbf{0}, \mathbf{w}_0$;
**for** $t = 0, 1, \ldots, T - 1$ **do**
    Randomly choose $B$ function indices, denoted as $\mathcal{I}_t$;
    Compute a mini-batch gradient $\mathbf{g}_t = \frac{1}{B} \sum_{i \in \mathcal{I}_t} \nabla f_i(\mathbf{w}_t)$;
    $\mathbf{u}_{t+1} = \beta \mathbf{u}_t + \frac{\mathbf{g}_t}{\|\mathbf{g}_t\|}$;
    $\mathbf{w}_{t+1} = \mathbf{w}_t - \eta \mathbf{u}_{t+1}$;
**end for**

---

Since $\eta \leq (1 - \beta)^2 / ((1 + \beta)L)$ is necessary for (4), we firstly obtain that $B \leq \mathcal{O}(\sqrt{\mathcal{C}}/L)$. Furthermore, according to the right term of (5), we have to set $B$ such that $B^2/\mathcal{C} \leq 1/\sqrt{\mathcal{C}}$, i.e., $B \leq \mathcal{C}^{1/4}$, for $\mathcal{O}(1/\epsilon^4)$ computation complexity guarantee. Hence in MSGD, we have to set the batch size satisfying

$$B \leq \mathcal{O}(\min\{\frac{\sqrt{\mathcal{C}}}{L}, \mathcal{C}^{1/4}\}). \tag{6}$$

We can observe that a larger $L$ leads to a smaller batch size in MSGD. If $B$ does not satisfy (6), MSGD will get higher computation complexity.

In fact, to the best of our knowledge, among all the existing convergence analysis of SGD and its variants on both convex and non-convex problems, we can observe three necessary conditions for the $\mathcal{O}(1/\epsilon^4)$ computation complexity guarantee Li et al. (2014b;a); Lian et al. (2017); Yu et al. (2019b;a): (a) the objective function is $L$-smooth; (b) the learning rate $\eta$ is less than $\mathcal{O}(1/L)$; (c) the batch size $B$ is proportional to the learning rate $\eta$. One direct corollary is that the batch size is limited by the smooth constant $L$, i.e., $B \leq \mathcal{O}(1/L)$. Hence, we can not increase the batch size casually in these SGD based methods. Otherwise, it may slow down the convergence rate and we need to compute more gradients, which is consistent with the observations in Hoffer et al. (2017).

## 4 STOCHASTIC NORMALIZED GRADIENT DESCENT WITH MOMENTUM

In this section, we propose our novel methods, called stochastic normalized gradient descent with momentum (SNGM), which is presented in Algorithm 1. In the $t$-th iteration, SNGM runs the following update:

$$\mathbf{u}_{t+1} = \beta \mathbf{u}_t + \frac{\mathbf{g}_t}{\|\mathbf{g}_t\|}, \tag{7}$$

$$\mathbf{w}_{t+1} = \mathbf{w}_t - \eta \mathbf{u}_{t+1}, \tag{8}$$

where $\mathbf{g}_t = 1/B \sum_{i \in \mathcal{I}_t} \nabla f_i(\mathbf{w}_t)$ is a stochastic mini-batch gradient with a batch size of $B$. When $\beta = 0$, SNGM will degenerate to stochastic normalized gradient descent (SNGD) Hazan et al. (2015). The $\mathbf{u}_t$ is a variant of Polyak's momentum. But different from Polyak's MSGD which adopts $\mathbf{g}_t$ directly for updating $\mathbf{u}_{t+1}$, SNGM adopts the normalized gradient $\mathbf{g}_t/\|\mathbf{g}_t\|$ for updating $\mathbf{u}_{t+1}$. In MSGD, we can observe that if $\mathbf{g}_t$ is large, then $\mathbf{u}_t$ may be large as well and this may lead to a bad model parameter. Hence, we have to control the learning rate in MSGD, i.e., $\eta \leq (1/L)$, for a $L$-smooth objective function. The following lemma shows that $\|\mathbf{u}_t\|$ in SNGM can be well controlled whatever $\mathbf{g}_t$ is large or small.

**Lemma 2** *Let $\{\mathbf{u}_t\}$ be the sequence produced by (7), then we have $\forall t \geq 0$,*

$$\|\mathbf{u}_t\| \leq \frac{1}{1 - \beta}.$$

### 4.1 SMOOTH OBJECTIVE FUNCTION

For a smooth objective function, we have the following convergence result of SNGM:

Table 1: Comparison between MSGD and SNGM for a $L$-smooth objective function. $\mathcal{C}$ denotes the computation complexity (total number of gradient computation).

|  | $\frac{1}{T}\sum_{t=0}^{T-1}\mathbb{E}\|\nabla F(\mathbf{w}_t)\|$ | learning rate | batch size |
|---|---|---|---|
| MSGD | $\sqrt{\mathcal{O}(\frac{1}{\sqrt{\mathcal{C}}}) + \mathcal{O}(\frac{B^2}{\mathcal{C}})}$ | $\frac{B}{\sqrt{\mathcal{C}}}$ | $\min\{\frac{\sqrt{\mathcal{C}}}{L}, \mathcal{C}^{1/4}\}$ |
| SNGM | $\mathcal{O}(\frac{1}{\mathcal{C}^{1/4}})$ | $\frac{\sqrt{B}}{\sqrt{\mathcal{C}}}$ | $\sqrt{\mathcal{C}}$ |

**Theorem 1** *Let $F(\mathbf{w})$ be a $L$-smooth function ($L > 0$). The sequence $\{\mathbf{w}_t\}$ is produced by Algorithm 1. Then for any $\eta > 0, B > 0$, we have*

$$\frac{1}{T}\sum_{t=0}^{T-1}\mathbb{E}\|\nabla F(\mathbf{w}_t)\| \leq \frac{2(1-\beta)[F(\mathbf{w}_0) - F(\mathbf{w}^*)]}{\eta T} + L\kappa\eta + \frac{2\sigma}{\sqrt{B}}, \tag{9}$$

*where $\kappa = \frac{1+\beta}{(1-\beta)^2}$.*

**Proof 1** *See the supplementary.*

We can observe that different from (4) which needs $\eta \leq \mathcal{O}(1/L)$, (9) is true for any positive learning rate. According to Theorem 1, we obtain the following computation complexity of SNGM:

**Corollary 1** *Let $F(\mathbf{w})$ be a $L$-smooth function ($L > 0$). The sequence $\{\mathbf{w}_t\}$ is produced by Algorithm 1. Given any total number of gradient computation $\mathcal{C} > 0$, let $T = \lceil \mathcal{C}/B \rceil$,*

$$B = \sqrt{\frac{\mathcal{C}(1-\beta)\sigma^2}{2L(1+\beta)(F(\mathbf{w}_0) - F(\mathbf{w}^*))}},$$

*and*

$$\eta = \sqrt{\frac{2(1-\beta)^3(F(\mathbf{w}_0) - F(\mathbf{w}^*))B}{(1+\beta)L\mathcal{C}}}.$$

*Then we have*

$$\frac{1}{T}\sum_{t=0}^{T-1}\mathbb{E}\|\nabla F(\mathbf{w}_t)\| \leq 2\sqrt{2}\sqrt[4]{\frac{8L(1+\beta)[F(\mathbf{w}_0) - F(\mathbf{w}^*)]\sigma^2}{(1-\beta)\mathcal{C}}} = \mathcal{O}(\frac{1}{\mathcal{C}^{1/4}}).$$

*Hence, the computation complexity for achieving an $\epsilon$-stationary point is $\mathcal{O}(1/\epsilon^4)$.*

It is easy to verify that the $\eta$ and $B$ in Corollary 1 make the right term of (9) minimal. However, the $\eta$ and $B$ rely on the $L$ and $F(\mathbf{w}^*)$ which are usually unknown in practice. The following corollary shows the computation complexity of SNGM with simple settings about learning rate and batch size.

**Corollary 2** *Let $F(\mathbf{w})$ be a $L$-smooth function ($L > 0$). The sequence $\{\mathbf{w}_t\}$ is produced by Algorithm 1. Given any total number of gradient computation $\mathcal{C} > 0$, let $T = \lceil \mathcal{C}/B \rceil$, $B = \sqrt{\mathcal{C}}$ and $\eta = \sqrt{B/\mathcal{C}}$. Then we have*

$$\frac{1}{T}\sum_{t=0}^{T-1}\mathbb{E}\|\nabla F(\mathbf{w}_t)\| \leq \frac{2(1-\beta)[F(\mathbf{w}_0) - F(\mathbf{w}^*)]}{\mathcal{C}^{1/4}} + \frac{L(1+\beta)}{(1-\beta)^2\mathcal{C}^{1/4}} + \frac{2\sigma}{\mathcal{C}^{1/4}} = \mathcal{O}(\frac{1}{\mathcal{C}^{1/4}}).$$

*Hence, the computation complexity for achieving an $\epsilon$-stationary point is $\mathcal{O}(1/\epsilon^4)$.*

According to Corollary 2, the batch size of SNGM can be set as $\mathcal{O}(\sqrt{\mathcal{C}})$, which does not rely on the smooth constant $L$, and the $\mathcal{O}(1/\epsilon^4)$ computation complexity is still guaranteed (see Table 1). Hence, SNGM can adopt a larger batch size than MSGD, especially when $L$ is large.

## 4.2 Relaxed Smooth Objective Function

Recently, the authors in Zhang et al. (2020) observe the relaxed smooth property in deep neural networks. According to Definition 2, the relaxed smooth property is more general than $L$-smooth property. For a relaxed smooth objective function, we have the following convergence result of SNGM:

**Theorem 2** *Let $F(\mathbf{w})$ be a $(L, \lambda)$-smooth function ($L \geq 0, \lambda > 0$). The sequence $\{\mathbf{w}_t\}$ is produced by Algorithm 1 with the learning rate $\eta$ and batch size $B$. Then we have*

$$\frac{1}{T} \sum_{t=0}^{T-1} \mathbb{E}\|\nabla F(\mathbf{w}_t)\| \leq \frac{2(1-\beta)[F(\mathbf{w}_0) - F(\mathbf{w}^*)]}{\eta T} + 8L\kappa\eta + \frac{4\sigma}{\sqrt{B}}, \tag{10}$$

*where $\kappa = \frac{1+\beta}{(1-\beta)^2}$ and $\eta \leq 1/(8\kappa\lambda)$.*

**Proof 2** *The proof is similar to that of Theorem 1. See the supplementary.*

According to Theorem 2, we obtain the computation complexity of SNGM:

**Corollary 3** *Let $F(\mathbf{w})$ be a $(L, \lambda)$-smooth function ($L \geq 0, \lambda \geq 0$). The sequence $\{\mathbf{w}_t\}$ is produced by Algorithm 1. Given any total number of gradient computation $\mathcal{C} > 0$, let $T = \lceil \mathcal{C}/B \rceil$, $B = \sqrt{\mathcal{C}}$ and $\eta = \sqrt[4]{1/\mathcal{C}} \leq 1/(8\kappa\lambda)$. Then we have*

$$\frac{1}{T} \sum_{t=0}^{T-1} \mathbb{E}\|\nabla F(\mathbf{w}_t)\| \leq \frac{2(1-\beta)[F(\mathbf{w}_0) - F(\mathbf{w}^*)]}{\mathcal{C}^{1/4}} + \frac{8L(1+\beta)}{(1-\beta)^2\mathcal{C}^{1/4}} + \frac{4\sigma}{\mathcal{C}^{1/4}} = \mathcal{O}(\frac{1}{\mathcal{C}^{1/4}}).$$

*Hence, the computation complexity for achieving an $\epsilon$-stationary point is $\mathcal{O}(1/\epsilon^4)$.*

According to Corollary 3, SNGM with a batch size of $B = \sqrt{\mathcal{C}}$ can still guarantee a $\mathcal{O}(1/\epsilon^4)$ computation complexity for a relaxed smooth objective function.

## 5 Experiments

All experiments are conducted with the platform of PyTorch, on a server with eight NVIDIA Tesla V100 (32G) GPU cards. The datasets for evaluation include CIFAR10 and ImageNet.

### 5.1 On CIFAR10

First, we evaluate SNGM by training ResNet20 and ResNet56 on CIFAR10. CIFAR10 contains 50k training samples and 10k test samples. We compare SNGM with MSGD and an existing large batch training method LARS You et al. (2017). We implement LARS by using the open source code [1]. The standard strategy He et al. (2016) for training the two models on CIFAR10 is using MSGD with a weight decay of 0.0001, a batch size of 128, an initial learning rate of 0.1, and dividing the learning rate at the 80th and 120th epochs. We also adopt this strategy for MSGD in this experiment. For SNGM and LARS, we set a large batch size of 4096 and also a weight decay of 0.0001. Following You et al. (2017), we adopt the poly power learning rate strategy and adopt the gradient accumulation Ott et al. (2018) with a batch size of 128 for the two large batch training methods. The momentum coefficient is 0.9 for all methods. Different from existing heuristic methods for large batch training, we do not adopt the warm-up strategy for SNGM.

The results are presented in Figure 2. As can be seen, SNGM achieves better convergence rate on training loss than LARS. The detailed information about the final convergence results is presented in Table 2. We can observe that MSGD with a batch size of 4096 leads to a significant drop of test accuracy. SNGM with a batch size of 4096 achieves almost the same test accuracy as MSGD with a batch size of 128. But the other large batch training method LARS achieves worse test accuracy than MSGD with a batch size of 128. These results successfully verify the effectiveness of SNGM.

---

[1] https://github.com/noahgolmant/pytorch-lars

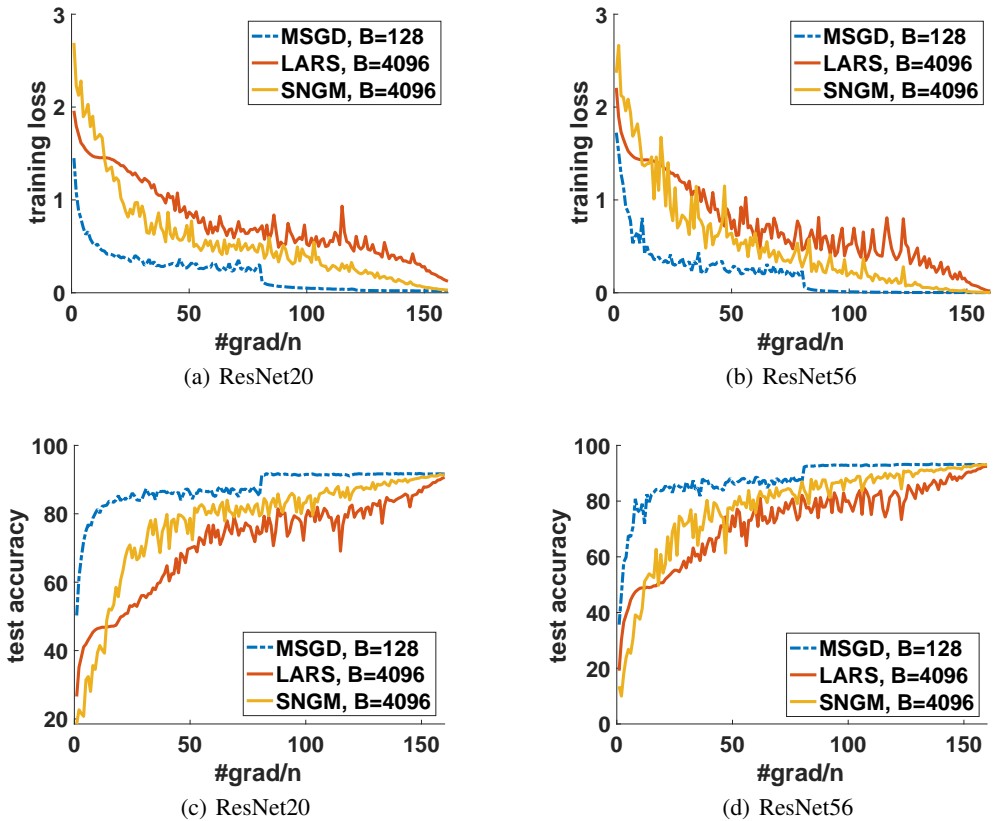

Figure 2: Learning curves on CIFAR10.

Table 2: Experimental results on CIFAR10. In LARS with warm-up, we adopt the gradual warm-up strategy and a power of 2, which is the same setting as that in You et al. (2017). In the warm-up stage (5 epochs), the learning rate increases from 0.1 to the target (2.4 in ResNet20 and 2.8 in ResNet56) gradually.

|  |  | warm-up | initial learning rate | power | batch size | test accuracy |
|---|---|---|---|---|---|---|
| ResNet20 | MSGD | - | 0.1 | - | 128 | 91.63% |
|  | MSGD | - | 0.4 | - | 4096 | 89.25% |
|  | LARS | No | 0.8 | 1.1 | 4096 | 90.66% |
|  | LARS | Yes | 2.4 | 2 | 4096 | 90.80% |
|  | SNGM | No | 1.6 | 1.1 | 4096 | 91.42% |
| ResNet56 | MSGD | - | 0.1 | - | 128 | 93.11% |
|  | MSGD | - | 0.3 | - | 4096 | 88.55% |
|  | LARS | No | 0.64 | 1.1 | 4096 | 92.46% |
|  | LARS | Yes | 2.8 | 2 | 4096 | 92.98% |
|  | SNGM | No | 1.3 | 1.1 | 4096 | 93.12% |

## 5.2 ON IMAGENET

We also compare SNGM with MSGD by training ResNet18 and ResNet50 on ImageNet. The standard strategy He et al. (2016) for training the two models on ImageNet is using MSGD with a weight decay of 0.0001, a batch size of 256, an initial learning rate of 0.1, and dividing the learning rate at the 30th and 60th epochs. We also adopt this strategy for MSGD in this experiment. For SNGM, we set a larger batch size of 8192 and a weight decay of 0.0001. We still adopt the poly power learning rate and the gradient accumulation with a batch size of 128 for SNGM. We do not adopt the warm-up strategy for SNGM either. The momentum coefficient is 0.9 in the two methods. The results are

presented in Figure 3 and Table 3. As can be seen, SNGM with a larger batch size achieves almost the same test accuracy as MSGD with a small batch size.

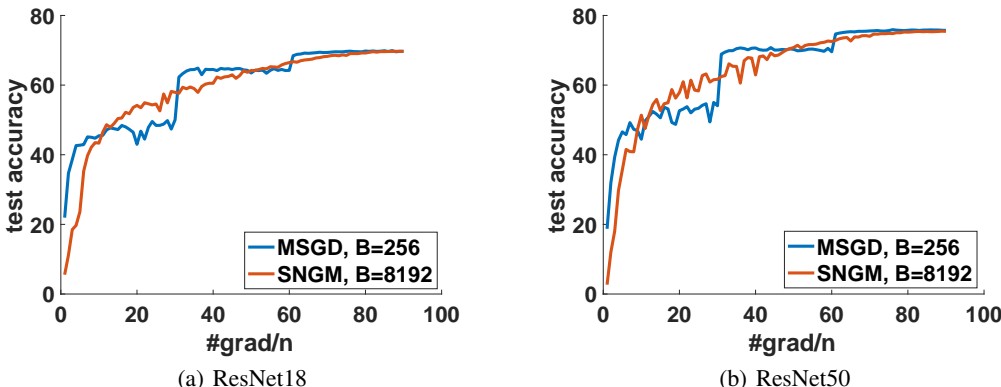

(a) ResNet18               (b) ResNet50

Figure 3: Learning curves on ImageNet.

Table 3: Experimental results on ImageNet.

|  |  | initial learning rate | power | batch size | test accuracy |
|---|---|---|---|---|---|
| ResNet18 | MSGD | 0.1 | - | 256 | 69.71% |
|  | SNGM | 0.8 | 2 | 8192 | 69.65% |
| ResNet50 | MSGD | 0.1 | - | 256 | 75.70% |
|  | SNGM | 0.8 | 2 | 8192 | 75.42% |

## 6 CONCLUSION

In this paper, we propose a novel method called stochastic normalized gradient descent with momentum (SNGM), for large batch training. We theoretically prove that compared to MSGD which is one of the most widely used variants of SGD, SNGM can adopt a larger batch size to converge to the $\epsilon$-stationary point with the same computation complexity. Empirical results on deep learning also show that SNGM can achieve the state-of-the-art accuracy with a large batch size.

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

# A  APPENDIX

## A.1  PROOF OF LEMMA 1

The proof follows Zhang et al. (2020). We put it here for completeness. For any $\mathbf{u}, \mathbf{w}$, let $r(x) = x(\mathbf{u} - \mathbf{w}) + \mathbf{w}$, $p(x) = \|\nabla\phi(r(x))\|$, $x \in [0,1]$. Then we have

$$
\begin{aligned}
p(x) =& \|\nabla\phi(r(x))\| = \|\int_0^x H_\phi(r(y))r'(y)dy + \nabla\phi(r(0))\| \\
=& \|\int_0^x H_\phi(r(y))(\mathbf{u} - \mathbf{w})dy + \nabla\phi(\mathbf{w})\| \\
\leq& \|\mathbf{u} - \mathbf{w}\| \int_0^x \|H_\phi(r(y))\|dy + \|\nabla\phi(\mathbf{w})\| \\
\leq& \alpha \int_0^x (L + \lambda\|\nabla\phi(r(y))\|)dy + \|\nabla\phi(\mathbf{w})\| \\
=& L\alpha + \|\nabla\phi(\mathbf{w})\| + \lambda\alpha \int_0^x p(y)dy.
\end{aligned}
$$

According to Gronwall's Inequality, we obtain

$$
p(x) \leq (L\alpha + \|\nabla\phi(\mathbf{w})\|)e^{\lambda\alpha}.
$$

## A.2  PROOF OF LEMMA 2

According to (7), we have

$$
\begin{aligned}
\|\mathbf{u}_{t+1}\| \leq& \beta\|\mathbf{u}_t\| + 1 \\
\leq& \beta^2\|\mathbf{u}_{t-1}\| + \beta + 1 \\
\leq& \beta^{t+1}\|\mathbf{u}_0\| + \beta^t + \beta^{t-1} + \cdots + 1 \\
\leq& \frac{1}{1-\beta}.
\end{aligned}
$$

## A.3  PROOF OF THEOREM 1

Let $\mathbf{z}_t = \mathbf{w}_t + \frac{\beta}{1-\beta}(\mathbf{w}_t - \mathbf{w}_{t-1})$, then we have $\mathbf{w}_{t+1} = \mathbf{w}_t - \eta\frac{\mathbf{g}_t}{\|\mathbf{g}_t\|} + \beta(\mathbf{w}_t - \mathbf{w}_{t-1})$ and

$$
\begin{aligned}
\mathbf{z}_{t+1} =& \frac{1}{1-\beta}\mathbf{w}_{t+1} - \frac{\beta}{1-\beta}\mathbf{w}_t \\
=& \mathbf{z}_t - \frac{\eta}{1-\beta}\frac{\mathbf{g}_t}{\|\mathbf{g}_t\|}.
\end{aligned}
$$

Using the smooth property, we obtain

$$
\begin{aligned}
F(\mathbf{z}_{t+1}) \leq& F(\mathbf{z}_t) - \frac{\eta}{1-\beta}\nabla F(\mathbf{z}_t)^T\frac{\mathbf{g}_t}{\|\mathbf{g}_t\|} + \frac{L\eta^2}{2(1-\beta)^2} \\
=& F(\mathbf{z}_t) - \frac{\eta}{1-\beta}\|\mathbf{g}_t\| + \frac{L\eta^2}{2(1-\beta)^2} \\
& - \frac{\eta}{1-\beta}[(\nabla F(\mathbf{z}_t) - \nabla F(\mathbf{w}_t))^T\frac{\mathbf{g}_t}{\|\mathbf{g}_t\|} + (\nabla F(\mathbf{w}_t) - \mathbf{g}_t)^T\frac{\mathbf{g}_t}{\|\mathbf{g}_t\|}] \\
\leq& F(\mathbf{z}_t) - \frac{\eta}{1-\beta}\|\mathbf{g}_t\| + \frac{L\eta^2}{2(1-\beta)^2} + \frac{\eta}{1-\beta}[L\|\mathbf{z}_t - \mathbf{w}_t\| + \|\nabla F(\mathbf{w}_t) - \mathbf{g}_t\|] \quad (11)
\end{aligned}
$$

Since $\mathbf{w}_{t+1} - \mathbf{w}_t = \beta(\mathbf{w}_t - \mathbf{w}_{t-1}) - \eta\mathbf{g}_t/\|\mathbf{g}_t\|$, we obtain

$$
\|\mathbf{w}_{t+1} - \mathbf{w}_t\| \leq \beta\|\mathbf{w}_t - \mathbf{w}_{t-1}\| + \eta \leq \frac{\eta}{1-\beta}.
$$

Hence, $\|\mathbf{w}_t - \mathbf{w}_{t-1}\| \le \eta/(1-\beta)$ and

$$\|\mathbf{z}_t - \mathbf{w}_t\| = \frac{\beta}{1-\beta}\|\mathbf{w}_t - \mathbf{w}_{t-1}\| \le \frac{\beta\eta}{(1-\beta)^2}. \tag{12}$$

Combining the above equations, we obtain

$$\|\mathbf{g}_t\| \le \frac{(1-\beta)[F(\mathbf{z}_t) - F(\mathbf{z}_{t+1})]}{\eta} + \frac{L\eta}{2(1-\beta)} + \frac{L\beta\eta}{(1-\beta)^2} + \|\nabla F(\mathbf{w}_t) - \mathbf{g}_t\|.$$

Since $\|\nabla F(\mathbf{w}_t)\| \le \|\nabla F(\mathbf{w}_t) - \mathbf{g}_t\| + \|\mathbf{g}_t\|$, we obtain

$$\|\nabla F(\mathbf{w}_t)\| \le \frac{(1-\beta)[F(\mathbf{z}_t) - F(\mathbf{z}_{t+1})]}{\eta} + \frac{L\eta}{2(1-\beta)} + \frac{L\beta\eta}{(1-\beta)^2} + 2\|\nabla F(\mathbf{w}_t) - \mathbf{g}_t\|.$$

Using the fact that $\mathbb{E}\|\nabla F(\mathbf{w}_t) - \mathbf{g}_t\| \le \sigma/\sqrt{B}$ and summing up the above inequality from $t = 0$ to $T - 1$, we obtain

$$\frac{1}{T}\sum_{t=0}^{T-1} \mathbb{E}\|\nabla F(\mathbf{w}_t)\| \le \frac{2(1-\beta)[F(\mathbf{w}_0) - F(\mathbf{w}^*)]}{\eta T} + L\kappa\eta + \frac{2\sigma}{\sqrt{B}}.$$

### A.4 PROOF OF THEOREM 2

Let $\mathbf{z}_t = \mathbf{w}_t + \frac{\beta}{1-\beta}(\mathbf{w}_t - \mathbf{w}_{t-1})$, then we have $\mathbf{w}_{t+1} = \mathbf{w}_t - \eta\frac{\mathbf{g}_t}{\|\mathbf{g}_t\|} + \beta(\mathbf{w}_t - \mathbf{w}_{t-1})$ and

$$\begin{aligned}
\mathbf{z}_{t+1} &= \frac{1}{1-\beta}\mathbf{w}_{t+1} - \frac{\beta}{1-\beta}\mathbf{w}_t \\
&= \frac{1}{1-\beta}[\mathbf{w}_t - \eta\frac{\mathbf{g}_t}{\|\mathbf{g}_t\|} + \beta(\mathbf{w}_t - \mathbf{w}_{t-1})] - \frac{\beta}{1-\beta}\mathbf{w}_t \\
&= \frac{1}{1-\beta}\mathbf{w}_t - \frac{\beta}{1-\beta}\mathbf{w}_{t-1} - \frac{\eta}{1-\beta}\frac{\mathbf{g}_t}{\|\mathbf{g}_t\|} \\
&= \mathbf{z}_t - \frac{\eta}{1-\beta}\frac{\mathbf{g}_t}{\|\mathbf{g}_t\|}.
\end{aligned}$$

Using the Taylor theorem, there exists $\xi_t$ such that

$$\begin{aligned}
F(\mathbf{z}_{t+1}) &\le F(\mathbf{z}_t) - \frac{\eta}{1-\beta}\nabla F(\mathbf{z}_t)^T\frac{\mathbf{g}_t}{\|\mathbf{g}_t\|} + \frac{\|H_F(\xi_t)\|\eta^2}{2(1-\beta)^2} \\
&= F(\mathbf{z}_t) - \frac{\eta}{1-\beta}\|\mathbf{g}_t\| + \frac{\|H_F(\xi_t)\|\eta^2}{2(1-\beta)^2} \\
&\quad - \frac{\eta}{1-\beta}[(\nabla F(\mathbf{z}_t) - \nabla F(\mathbf{w}_t))^T\frac{\mathbf{g}_t}{\|\mathbf{g}_t\|} + (\nabla F(\mathbf{w}_t) - \mathbf{g}_t)^T\frac{\mathbf{g}_t}{\|\mathbf{g}_t\|}].
\end{aligned} \tag{13}$$

Let $\psi_t(\mathbf{w}) = (\nabla F(\mathbf{w}) - \nabla F(\mathbf{w}_t))^T\frac{\mathbf{g}_t}{\|\mathbf{g}_t\|}$. Using the Taylor theorem, there exists $\zeta_t$ such that

$$\begin{aligned}
|\psi_t(\mathbf{z}_t)| &= |\psi_t(\mathbf{w}_t) + \nabla\psi_t(\zeta_t)(\mathbf{z}_t - \mathbf{w}_t)| = |\nabla\psi(\zeta_t)(\mathbf{z}_t - \mathbf{w}_t)| \\
&\le \|H_F(\zeta_t)\|\|\mathbf{z}_t - \mathbf{w}_t\|.
\end{aligned} \tag{14}$$

Combining (13) and (14), we obtain

$$\begin{aligned}
\|\mathbf{g}_t\| &\le \frac{(1-\beta)[F(\mathbf{z}_t) - F(\mathbf{z}_{t+1})]}{\eta} + \frac{\|H_F(\xi_t)\|\eta}{2(1-\beta)} \\
&\quad + (\|H_F(\zeta_t)\|\|\mathbf{z}_t - \mathbf{w}_t\| + \|\nabla F(\mathbf{w}_t) - \mathbf{g}_t\|).
\end{aligned} \tag{15}$$

Since $\mathbf{w}_{t+1} - \mathbf{w}_t = \beta(\mathbf{w}_t - \mathbf{w}_{t-1}) - \eta\mathbf{g}_t/\|\mathbf{g}_t\|$, we obtain

$$\|\mathbf{w}_{t+1} - \mathbf{w}_t\| \le \beta\|\mathbf{w}_t - \mathbf{w}_{t-1}\| + \eta \le \frac{\eta}{1-\beta}.$$

Hence, $\|\mathbf{w}_t - \mathbf{w}_{t-1}\| \leq \eta/(1-\beta)$ and

$$\|\mathbf{z}_t - \mathbf{w}_t\| = \frac{\beta}{1-\beta}\|\mathbf{w}_t - \mathbf{w}_{t-1}\| \leq \frac{\beta\eta}{(1-\beta)^2}. \tag{16}$$

Combining (15) and (16), we obtain

$$\|\mathbf{g}_t\| \leq \frac{(1-\beta)[F(\mathbf{z}_t) - F(\mathbf{z}_{t+1})]}{\eta} + \frac{\|H_F(\xi_t)\|\eta}{2(1-\beta)} + \frac{\|H_F(\zeta_t)\|\beta\eta}{(1-\beta)^2}$$
$$+ \|\nabla F(\mathbf{w}_t) - \mathbf{g}_t\|.$$

Since $\|\nabla F(\mathbf{w}_t)\| \leq \|\nabla F(\mathbf{w}_t) - \mathbf{g}_t\| + \|\mathbf{g}_t\|$, we obtain

$$\|\nabla F(\mathbf{w}_t)\| \leq \frac{(1-\beta)[F(\mathbf{z}_t) - F(\mathbf{z}_{t+1})]}{\eta} + \frac{\eta}{2(1-\beta)}\|H_F(\xi_t)\| + \frac{\beta\eta}{(1-\beta)^2}\|H_F(\zeta_t)\|$$
$$+ 2\|\nabla F(\mathbf{w}_t) - \mathbf{g}_t\|.$$

Next, we bound the two Hessian matrices. For convenience, we denote $\kappa = \frac{1+\beta}{(1-\beta)^2}$. Since $\|\mathbf{z}_t - \mathbf{w}_t\| \leq \beta\eta/(1-\beta)^2$ and

$$\|\mathbf{z}_{t+1} - \mathbf{w}_t\| \leq \|\mathbf{z}_{t+1} - \mathbf{z}_t\| + \|\mathbf{z}_t - \mathbf{w}_t\|$$
$$\leq \eta\left(\frac{1}{1-\beta} + \frac{\beta}{(1-\beta)^2}\right)$$
$$\leq \kappa\eta$$
$$\leq \frac{1}{\lambda},$$

we obtain

$$\|H_F(\zeta_t)\| \leq L + (L + \lambda\|\nabla F(\mathbf{w}_t)\|)e,$$
$$\|H_F(\xi_t)\| \leq L + (L + \lambda\|\nabla F(\mathbf{w}_t)\|)e.$$

Then we obtain

$$\|\nabla F(\mathbf{w}_t)\| \leq \frac{(1-\beta)[F(\mathbf{z}_t) - F(\mathbf{z}_{t+1})]}{\eta} + \left[\frac{\eta}{2(1-\beta)} + \frac{\beta\eta}{(1-\beta)^2}\right][L + (L + \lambda\|\nabla F(\mathbf{w}_t)\|)e]$$
$$+ 2\|\nabla F(\mathbf{w}_t) - \mathbf{g}_t\|$$
$$\leq \frac{(1-\beta)[F(\mathbf{z}_t) - F(\mathbf{z}_{t+1})]}{\eta} + 4\kappa\eta[L + \lambda\|\nabla F(\mathbf{w}_t)\|]$$
$$+ 2\|\nabla F(\mathbf{w}_t) - \mathbf{g}_t\|.$$

Since $4\lambda\kappa\eta \leq 1/2$, we obtain

$$\|\nabla F(\mathbf{w}_t)\| \leq \frac{2(1-\beta)[F(\mathbf{z}_t) - F(\mathbf{z}_{t+1})]}{\eta} + 8Lc\eta + 4\|\nabla F(\mathbf{w}_t) - \mathbf{g}_t\|.$$

Summing up the above inequality from $t = 0$ to $T - 1$, we obtain

$$\frac{1}{T}\sum_{t=0}^{T-1}\mathbb{E}\|\nabla F(\mathbf{w}_t)\| \leq \frac{2(1-\beta)[F(\mathbf{w}_0) - F(\mathbf{w}^*)]}{\eta T} + 8L\kappa\eta + \frac{4\sigma}{\sqrt{B}}.$$

where $\eta \leq \frac{1}{8\lambda\kappa}$ and we use the fact that $\mathbb{E}\|\nabla F(\mathbf{w}_t) - \mathbf{g}_t\| \leq \sigma/\sqrt{B}$.

## A.5 PROOF OF COROLLARY 1

Let $x = 2(1-\beta)[F(\mathbf{w}_0) - F(\mathbf{w}^*)]$, $y = L\kappa$, $z = 2\sigma$. Then we have

$$\frac{xB}{\mathcal{C}\eta} + y\eta + \frac{z}{\sqrt{B}} \geq 2\sqrt{\frac{xyB}{\mathcal{C}}} + \frac{z}{\sqrt{B}} \geq 2\sqrt{2z\sqrt{\frac{xyB}{\mathcal{C}}}} = 2\sqrt{2}\sqrt[4]{\frac{xyz^2}{\mathcal{C}}}.$$

The equal sign works if and only if $\eta = \sqrt{Bx/\mathcal{C}y}$, $B = \sqrt{\mathcal{C}z^2/(4xy)}$. Then we obtain

$$\frac{1}{T}\sum_{t=0}^{T-1}\mathbb{E}\|\nabla F(\mathbf{w}_t)\| \leq 2\sqrt{2}\sqrt[4]{\frac{8L(1+\beta)[F(\mathbf{w}_0) - F(\mathbf{w}^*)]\sigma^2}{(1-\beta)\mathcal{C}}}.$$

### A.6 PROOF OF COROLLARY 2

By plugging $T = \lceil \mathcal{C}/B \rceil$, $B = \sqrt{\mathcal{C}}$ and $\eta = \sqrt[4]{1/\mathcal{C}}$ into (9), we obtain the result.

### A.7 PROOF OF COROLLARY 3

By plugging $T = \lceil \mathcal{C}/B \rceil$, $B = \sqrt{\mathcal{C}}$ and $\eta = \sqrt[4]{1/\mathcal{C}}$ into (10), we obtain the result.

