# OpenReview forum: "Stochastic Normalized Gradient Descent with Momentum for Large Batch Training"
_ICLR.cc/2021/Conference — Reject_

### Official Review · AnonReviewer1 · 2020-10-13
**weak experimental evaluation**

**Rating:** 4
**Confidence:** 4

**Review:**

In this paper, the authors proposed an optimizer by the name of SNGM (stochastic normalized gradient descent with momentum). We can get the main idea by just reading its name. The authors claimed the main contributions are on the theoretical side. Overall, I think this paper is neat and clear. The figures and tables are easy to read and understand. The main idea is also highlighted in a proper way. I like the style of this paper.

Here are some of my concerns:

In terms of algorithm design, the idea of this paper is pretty simple: Momentum SGD plus normalized gradients. Similar ideas have been tried in many previous papers (e.g. [1]).

The authors claimed two theoretical contributions:

(1) SNGM is able to adopt a larger batch size to converge to the epsilon-stationary point with the same number of floating-point operations as SGD.

(2) They can prove that SNGM converges to an epsilon-stationary point with a computation complexity of O(1/epsilon^4).

I quickly went over the proof and did not find obvious flaws. However, because SNGM has a relatively simple algorithm structure, the theoretical study of this paper seems to be not a lot of hardcore work. The theoretical contribution is above the average but not very impressive.

The experimental results are not strong enough to convince potential users.

The authors used [3] as the LARS implementation. Please see also [4] to make sure the implementation is correct. The researchers in [4] [5] [6] reported a much higher accuracy than this paper by LARS optimizer in ImageNet/ResNet-50 training. If the authors can confirm the implementation is correct, then they need to make sure the hyper-parameters of all optimizers are properly tuned. Only in this way, we can have a fair comparison.

The authors used a batch size of 8192 in ImageNet training experiments.
According to previous papers like [2] [4] [5] [6], a batch size of 8192 is probably too small to be considered as a “large batch size”. It can be easier to evaluate the contribution of this paper if the authors can report the results using a batch size over 32K.
If the authors have out-of-memory issues with limited hardware resources, please check [7].

It will be better if the authors can provide a copy of the implementation with detailed hyper-parameters.

[1] https://arxiv.org/pdf/1707.04822v1.pdf

[2] https://openreview.net/pdf?id=Syx4wnEtvH

[3] https://github.com/noahgolmant/pytorch-lars

[4] https://arxiv.org/pdf/1909.09756.pdf

[5] https://arxiv.org/pdf/1807.11205.pdf

[6] https://arxiv.org/pdf/1811.06992.pdf

[7] https://discuss.pytorch.org/t/how-to-implement-accumulated-gradient/3822

---

### Official Review · AnonReviewer2 · 2020-10-27
**Review for Stochastic Normalized Gradient Descent with Momentum for Large Batch Training**

**Rating:** 4
**Confidence:** 4

**Review:**

###################################################################

Summary:

This paper proposes a new stochastic normalized gradient descent method with momentum (SNGM) for large batch training. They prove that unlike mometum SGD (MSGD), SNGM can adopt larger batch size to converge to the epsilon-stationary point with the same computation complexity (total number of gradient computation). The paper shows that SNGM with large batches is comparable to MSGD with small batches for training ResNet on CIFAR10 and ImageNet. The paper also shows that SNGM outperforms LARS on CIFAR10.

###################################################################

Reason for the Score:

Overall, I vote for rejecting. I have two main major concerns. First, the theoretical contributions of the paper are only small extensions from existing works and not significant. Second, very limited number of experiments are performed in the paper. The authors also do not compare the proposed method with popular works in large batch training that yield better results than those reported in the paper.

###################################################################

Strong points:

1. Analyzing the relationship between smoothness constant and batch size can potentially help provide a better understanding of large batch training algorithms.

2. The paper is well-written.

###################################################################

Weak points:

1. The analysis of the relationship between smoothness constant and batch size is a trivial extension from the results in Theorem 1 and Corollary 1 of Yu et al. (2019a).

2. The authors claim that existing works such as LARS lack theoretical evidence to explain why they can adopt a large batch size. However, theoretical explanations for normalized SGD with momentum like LARS are provided in Cutkosky et al. (2020).

3. The paper does not compare the proposed SNGM method with popular works in large batch training. In particular, in ImageNet experiments, the authors only compare the SNGM method with the baseline MSGD trained with small batches. In fact, when training ResNet50 on ImageNet with the batch size of 8192, LAMB proposed by You et al. (2020) reported the test accuracy of 76.66%, which is much better than the reported test accuracy of 75.42% attained by SNGM. Another existing work that reported a better result on the same task is Smith et al. (2018).

4. The main claim of the paper is that SNGM can adopt large batch size. However, only experiments with a batch size of 4096 on CIFAR and a batch size of  8192 on ImageNet are provided. More experiments with different batch sizes, especially larger ones, are needed.

###################################################################

Additional Concerns and Questions for the Authors:

1. LARS and LAMB also normalize the update in SGD. After reading the paper, I do not see the clear theoretical and practical advantages of SNGM over these methods.

2. The paper only provides experiments with ResNets and image datasets. Additional experiments on different models and tasks, such as training BERT for language modeling, are needed to validate the advantage of SNGM.

###################################################################

Minor Comments that did not Impact the Score:

1. It would be good if the authors can compare the runtime of SNGM with other methods.

2. If possible, error bars should be provided for the experimental results.

3. Related works should be discussed more.

###################################################################

References:

Hao Yu, Rong Jin, and Sen Yang. On the linear speedup analysis of communication efficient momentum SGD for distributed non-convex optimization. In Proceedings of the 36th International Conference on Machine Learning, 2019a.

Ashok Cutkosky and Harsh Mehta. Momentum improves normalized SGD. International Conference on Machine Learning, 2020.

Yang You, Jing Li, Sashank J. Reddi, Jonathan Hseu, Sanjiv Kumar, Srinadh Bhojanapalli, Xiaodan Song, James Demmel, Kurt Keutzer, and Cho-Jui Hsieh. Large batch optimization for deep learning: Training BERT in 76 minutes. In Proceedings of the International Conference on Learning Representations, 2020.

Smith, S., jan Kindermans, P., Ying, C., and Le, Q. V. Don’t decay the learning rate, increase the batch size. In International Conference on Learning Representations (ICLR), 2018.

###################################################################

Post Discussion Score:

The authors did not submit their rebuttal. After reading the comments from other reviewers, I decided to keep my original score for this paper.

---

### Official Review · AnonReviewer4 · 2020-10-28
**This paper theoretical convergence result of  a normalized version of MSGD in large batch training.**

**Rating:** 4
**Confidence:** 5

**Review:**

Summary:
Large batch training has been observed to not only significantly improve the training speed but also lead to a worse generalization performance. This paper considers how to improve the performance of MSGD in large batch training. They propose the so called normalized MSGD where instead of the gradient, the algorithm uses the normalized gradient to update the momentum. They also provide theoretical justification of this change by considering smooth and relaxed smooth function. O(1/\epsilon^4)  convergence rate is established.
#######################
Pros:

1. Though the idea of using normalized gradient to improve the algorithm’s  performance is not novel, there still lacks theoretical analysis. The paper successfully fills this gap.

2. The paper overall is well written and well structured. The authors give a clear path from the background to algorithm and then convergence analysis. I have no difficult in reading and understanding all the results.

#######################

Major Comments:

1. The algorithm considered in this paper is a simple combination of MSGD and gradient normalization. As such, none of the components are new. The theoretical results are not very surprising, and the techniques used are not new. I checked the proof, the only difference compared with MSGD’s proof is to bound the difference between gradient and normalized gradient which is not difficult given the smoothness condition (at least in my opinion). I may miss something and if I can be convinced on the difficulty of the analysis, I would like to raise the score.

2. The theoretical result only considers converging to a first order stationary point. However, in nonconvex setting, that is far away from claiming the algorithm can have better performance. I  understand the difficulty to prove second-order convergence, but at least some discussion should be added.

3. I suggest the authors to add more intuitive explanation to why this normalized MSGD works.

4. Numerically, the paper only compared with LARS which is a 2017 algorithm. I strongly suggest to add more experiments and compare SNGD with more benchmarks.


Overall, I vote for reject since I don’t find enough novel results here. I am willing to raise my score if the authors can convince me their contribution

---

### Official Review · AnonReviewer3 · 2020-10-28
**It seems that there are some issues in the theoretical results**

**Rating:** 3
**Confidence:** 3

**Review:**

This paper proposed a simple stochastic normalized gradient methods with momentum (SNGM) for non-convex optimization, the authors argued the main advantage of the method is it allows large-batch training. Theoretical convergence guarantees and  empirical results for neural network training are provided.

My main concerns are in the theoretical results: in the paper the authors argues the following: standard MSGD has convergence of O(/1/sqrt(C) + B^2/C), while SNGM, with B=sqrt(C), has convergence rate of O(1/C^(1/4)).  It seems to me it is unfair comparison as one has O(1/C^(1/2)) rate and the other has O(1/C^(1/4)) rate.

If we choose B=C^(1/4) in MSGD, then MSGD has convergence rate of O(1/C^(1/2)), which is much faster than the SNGM rate of O(1/C^(1/4)) ? Only if we fix the overall rate of O(1/C^(1/4)), then for MSGD the largest mini-batch size is B=C^(3/8), which is smaller than the SNGM minibatch size. Can SNGM obtain O(1/C^(1/2)) rate ? What is is mini--batch size limit under the fast O(1/C^(1/2)) rate ?

Please correct me if I am wrong.

---

### Decision · Program_Chairs · 2021-01-07
**Final Decision**

**Decision:**

Reject

**Comment:**

All reviewers agree that the contributions of this paper are not significant, and the paper does not compare well with many of the existing works. Authors did not respond.